# Reconnoitering the Role of Long-Noncoding RNAs in Hypertrophic Cardiomyopathy: A Descriptive Review

**DOI:** 10.3390/ijms22179378

**Published:** 2021-08-29

**Authors:** Syeda K. Shahzadi, Nerissa Naidoo, Alawi Alsheikh-Ali, Manfredi Rizzo, Ali A. Rizvi, Raul D. Santos, Yajnavalka Banerjee

**Affiliations:** 1Department of Basic Medical Sciences, Mohammed Bin Rashid University of Medicine and Health Sciences, Dubai 505055, United Arab Emirates; Syeda.Shahzadi@mbru.ac.ae (S.K.S.); alsheikhali@gmail.com (A.A.-A.); 2Dubai Health Authority, Dubai 66566, United Arab Emirates; 3Department of Health Promotion Sciences, Maternal and Infantile Care, Internal Medicine and Medical Specialties (PROMISE), University of Palermo, 90127 Palermo, Italy; manfredi.rizzo@unipa.it; 4Division of Endocrinology, Metabolism, and Lipids, School of Medicine, Emory University, Atlanta, GA 30322, USA; ali.abbas.rizvi@emory.edu; 5The Heart Institute, Faculty of Medicine, University of São Paulo, São Paulo 01000, Brazil; rauldsf@gmail.com; 6Centre of Medical Education, School of Medicine, University of Dundee, Dundee DD1 4HN, UK

**Keywords:** hypertrophic cardiomyopathy, long non-coding RNA, genetic variants, cardiovascular diseases

## Abstract

Hypertrophic cardiomyopathy (HCM) is the most common form of hereditary cardiomyopathy. It is characterized by an unexplained non-dilated hypertrophy of the left ventricle with a conserved or elevated ejection fraction. It is a genetically heterogeneous disease largely caused by variants of genes encoding for cardiac sarcomere proteins, including *MYH7*, *MYBPC3*, *ACTC1*, *TPM1*, *MYL2*, *MYL3*, *TNNI3*, and *TNNT23*. Preclinical evidence indicates that the enhanced calcium sensitivity of the myofilaments plays a key role in the pathophysiology of HCM. Notably, this is not always a direct consequence of sarcomeric variations but may also result from secondary mutation-driven alterations. Long non-coding RNAs (lncRNAs) are a large class of transcripts ≥200 nucleotides in length that do not encode proteins. Compared to coding mRNAs, most lncRNAs are not as well-annotated and their functions are greatly unexplored. Nevertheless, increasing evidence shows that lncRNAs are involved in a variety of biological processes and diseases including HCM. Accumulating evidence has indicated that lncRNAs are dysregulated in HCM, and closely related to sarcomere construction, calcium channeling and homeostasis of mitochondria. In this review, we have summarized the known regulatory and functional roles of lncRNAs in HCM.

## 1. Introduction

Hypertrophic cardiomyopathy (HCM) is one of the most prevalent inherited disorders of cardiomyocytes, without specific geographic, ethnic, or sex patterns of distribution. Studies estimate a prevalence of HCM at a range between 0.16–0.29% (~1:625 to 1:344 individuals) in the general adult population [1]. HCM is categorized as a typical single gene disorder with variable penetrance and expression, exhibiting an autosomal dominant pattern of inheritance. More than 1500 gene variants have been linked to HCM, with over 95% of them found in 11 or more genes encoding sarcomeric proteins, the heart’s contractile building blocks [2]. Variants of HCM with autosomal recessive and X-linked modes of inheritance have been described but are rare [3,4]. 

Phenotypically, HCM is characterized by an increase in the left ventricular wall thickness (end-diastolic left ventricular wall thickness ≥15 mm or the equivalent relative to the body surface area in children that cannot be solely explained by abnormal loading conditions (European Society of Cardiology Guidelines) [5]. Ventricular wall thickness to lesser degrees (13–14 mm) can also be diagnostic of HCM, especially when it is identified in family members [6]. Additionally, cardiac hypertrophy observed in HCM is typically asymmetric, with the greatest involvement most commonly at the basal interventricular septum subjacent to the aortic valve (Figure 1A). It is infrequently restricted to other myocardial regions, such as the posterior wall of the left ventricle in symmetric hypertrophy (Figure 1B) and at the apex in apical hypertrophy (Figure 1C).

Estimating the prevalence of HCM based on clinical detection of cardiac hypertrophy has many limitations. The first issue lies with age-dependent expression of cardiac hypertrophy, present in almost one-half and approximately three-fourths of patients with underlying contributing variants by the third and sixth decades of life, respectively. Due to the age-dependent expression of HCM variants, it is expected that its prevalence will be higher in the elderly. Attesting to this is the fact that HCM has been documented in 0.29% (1:333) of 60-year-old subjects undergoing echocardiography for cardiovascular evaluation in the United States, United Arab Emirates and in other parts of the globe [7,8,9,10]. Furthermore, when more sensitive imaging methods are used, or when more family members from a family diagnosed with HCM are evaluated clinically and through genetic testing, a much higher estimate of 0.6% (1:167) has been proposed [11,12,13]. This warrants that the mechanism underlying HCM be investigated at depth to identify novel biomarkers, allowing for an uncomplicated diagnosis of the condition. Accurate estimates can thus be obtained, facilitating the development of suitable therapeutic and management strategies.

Long non-coding RNAs (lncRNAs) have transcripts ≥200 nucleotides that do not code for proteins. In comparison to messenger (m) RNAs, lncRNAs are less well-annotated, and their role in the physiological milieu remains largely unexplored. Increasing evidence shows that lncRNAs are implicated in causation of HCM. However, a comprehensive review evaluating the link/association of lncRNA with HCM is currently absent in the literature. In this manuscript, we address this gap of information. First, we provide the reader with an overview of HCM, focusing on the molecular genetic aspects. Next, we elaborate on lncRNAs’ role in a variety of pathophysiological states focusing on HCM. Lastly, we identify areas of further research that still need to be addressed such that lncRNAs can be effectively employed as novel biomarkers for the early detection and suitable management of HCM.

## 2. Molecular Genetics of HCM

Based on several identified variants in major causal genes and all encoding sarcomere proteins, HCM is classified as a genetically heterogeneous condition. HCM patients are found to have some sort of genetic mutation in one-half of the cases on average [14,15]. Genetic dissection of HCM patients has revealed a series of alterations in more than 11 gene encoding sarcomeric proteins [16]. HCM follows the dominant autosomal inheritance pattern with variable penetrance and expression related to age or as a new variation in non-related family cases [17]. The most dominant mutation is a missense mutation, replacing one nucleic acid with another, resulting in the modification of the translated amino acid and subsequent protein. Deletions and insertions are also found to be common in the pathogenesis of HCM leading to the translation of a modified protein [18].

Most variations occur in genes that are responsible for normal functioning of contractile sarcomeric proteins: troponin I and T, myosin binding protein C, myosin heavy and light chains, α-actin, titin and α-tropomyosin. Nevertheless, in rare cases, variations in non-sarcomeric protein coding genes have also been reported in HCM patients [19]. The genes predominantly related to the HCM development are *TNNT2*, *MYH7*, *MYBPC3*, *ACTC1*, *TPMI*, *TNNI3*, *TNNC1*, *TNNC2*, *MYL2*, *MYL3*, *CSRP3*, and *MYOZ2* (Figure 2 and Table 1) [20].

The *MYH7* and *MYBPC3* (myosin-binding protein C) genes are the most prevalent and are recognized to be responsible for approximately 50% of patients with familial HCM [21,22]. Variants *TNNT2*, *TNNI3* (cardiac troponin I) and *TPM1* (α-tropomyosin) are comparatively infrequent causes of HCM and account for <10% of all cases [23,24]. Variations in *ACTC1* (cardiac α-actin), *MYL2* (myosin light chain 2), *MYL3* (myosin light chain 3) and *CSRP3* (cysteine and glycine-rich protein 3) are also found to cause HCM [25,26]. While variations in *TCAP* (telethonin) [27], *TTN* (titin) [28], *MYOZ2* (myozenin 2), *FHL1* (four-and-a-half LIM domains 1) [29], and *TRIM63* (ubiquitin E3 ligase tripartite protein 63 or MuRF1) usually occur in rare cases and small families [30].

The concept of phenocopies within the HCM setting is also vital to highlight. These patients might have a HCM phenotype without the reported HCM genetic variants, but often present are a few other diseases driving to a similar heart condition, such as LAMP2 cardiomyopathy, Fabry malady, PRKAG2, Pompe Disease, Noonan syndrome, Wolff-Parkinson-White syndrome, Barth syndrome, Forbes disease, and/or amyloidosis (Table 2) [31].

## 3. Molecular and Biological Functions of lncRNAs

Long non-coding RNAs (lncRNAs) are characterized as RNAs having a transcript length that surpasses 200 bp of nucleotides and are not decoded into proteins [32]. Although lncRNAs were once thought to be byproducts of RNA polymerase II transcripts without any specific biological function, an expansive collection of lncRNAs have been certified to modulate cellular processes, such as genome and chromosome and modification, nuclear transport and transcription stimulation, hence driving more analysts to investigate how lncRNAs impact human biology. Classification of lncRNAs can be done in terms of function, location, length, and mode of action and as of now there’s no single standard for their categorization. Agreeing to their position within the genome, they can be categorized as sense, antisense, bidirectional, intronic, intergenic, and enhancer lncRNAs [33]. At the same time, they are commonly sorted as bait, scaffold, signal, and guide lncRNAs according to their functional mechanisms [34]. Recent studies reported that lncRNAs can translate small peptides to fine-tune common processes in a tissue-specific way, further disclosing the esteem of lncRNAs as well as their complexity [35,36,37,38]. However, the modes in which lncRNAs modulate gene expression are complicated and therefore, have not been completely explained. LncRNAs can work through diverse modes of operations: (1) binding to transcription factors or DNA directly to attain gene expression control at the transcriptional level; (2) focusing on miRNAs, mRNAs, and proteins to regulate their activities at the posttranscriptional level and (3) intertwining with chromatin complexes to modulate epigenetic gene expression [39,40,41]. 

The mode of action of lncRNAs permits their presence in nearly all physiological processes in living cells, thus relating them to a broad range of diseases. LncRNAs have developed into potent novel molecules in disease prognosis, diagnosis and treatment. As a result of their crucial role in diseases, researchers are presently developing tools and technologies to make lncRNA-based drugs by targeting lncRNAs. The relationship between chronic diseases and lncRNAs is explicit. In any case, cancer is the foremost studied illness that is related to lncRNAs [42]. However, recent years has revealed accounting evidence that has emphasized the role of lncRNA modulation in cardiovascular development as well as in cardiovascular diseases [43]. A thorough functional characterization of lncRNAs may help in the monitoring, prevention and treatment of cardiovascular diseases, including cardiac hypertrophy, hypertension, myocardial infarction and heart failure. Recent studies show that lncRNA-DACHI modulates heart function, and it was upregulated in heart failure patients, whereas the knockdown of lncRNA-DACHI in Heart failure (HF) mice hampered the HF development [44]. LncRNA-Safe was found to promote myocardial fibrosis and its inhibition in fibroblasts impeded cardiac functions: showing that the Safe can be a novel target for antifibrotic therapy [45]. Until now, MHRT, CHRF, MALATA1 and many more lncRNAs have been investigated in the cardiovascular field, particularly in cardiac hypertrophy (Table 3) [33]. However, the functions of lncRNAs in diagnosis, prevention and treatment of HCM are still understudied and should be further explored to identify novel biomarkers for HCM.

## 4. Emerging Role of lncRNAs in Pathogenesis of HCM

Numerous studies demonstrate that lncRNAs regulate pathophysiological processes in cardiac hypertrophy by interacting with genes involved in mitochondria, sarcomeres, and calcium transition at the intracellular level (Table 3). Many review articles have been published on the recent advancement in establishing the role of lncRNA in physiological cardiac hypertrophy [76,77,78,79,80]. However, there is a dearth of research regarding the function of lncRNAs in the pathophysiology of hypertrophic cardiomyopathy (Table 4).

Current evidence suggests that lncRNAs could be a promising target for attenuating or even reversing the pathological phenotype of hypertrophic cardiomyopathy. To date, several animal experiments and bioinformatics analyses have been conducted to determine the possible role of specific lncRNAs in the production of hypertrophic cardiomyopathy. However, we focused on those studies only involving the known mechanism of action of lncRNAs in development of hypertrophic cardiomyopathy.

### 4.1. H19 Regulates HCM through miR-675

H19 is an lncRNA that is expressed in a variety of tissues and is significantly increased during myoblast differentiation and muscle regeneration. H19 may act as a sponge for several miRNAs, thereby regulating the expression of numerous genes and signaling pathways. In pathological heart failure and cardiac hypertrophy, H19 and its host miR-675 are upregulated. Some researchers identified H19 variants in a group of patients with hypertrophic cardiomyopathy and compared their allele and genotype frequencies to those of healthy controls. In patients without sarcomere variations, the H19 rs2107425 CC genotype was significantly increased (*p* = 0.017; odd ratio: 1.51). The H19 transcript sequence revealed heterozygous carriers of a rare allele, rs945977096 AG, that was absent in the controls. Their study established a statistically significant link between H19 variants and an increased risk of developing hypertrophic cardiomyopathy. In pathological HCM and heart failure tissues, H19 and its host miR-675 were upregulated. H19 silencing resulted in cardiomyocyte hypertrophy, while overexpression resulted in cell size reduction both at baseline and in response to phenylephrine, an effect that may be mediated by miR-675. The SNP rs2107425 was associated with HCM, with the C allele being significantly more prevalent in patients who did not have sarcomere variants. Thus, homozygotes for rs2107425 CC would have a higher risk of developing HCM [81].

### 4.2. LncRNA-MIAT Regulates HCM through miR-29a-3P

In GC cell lines and tissues, the miR-29a-3p is an important miRNA with poor expression that is regulated by myocardial infarction-associated transcript (MIAT). MIAT can suppress the expression of miR-29a-3p as an endogenous miRNA sponge by making bonds with the miR-29a-3p inside the GC cells. Many studies investigated the expression of MEG3, MIAT, and H19 in effort to identify the particular lncRNA that regulates the miR-29a expression in myocardial fibrosis. Negative correlation of MIAT with the miR-29a expression was discovered. Based on ROC analysis, miR-29a and MIAT may accurately predict the prognosis in HCM patients [82].

### 4.3. LncRNA-Mhrt Protects the Myocardium against HCM

Han et al., identified an important lncRNA transcript cluster from Myh7 loci, which presented a new heart failure mechanism linked with lncRNA-chromatin. They named this transcript as “Myosin Heavy Chain Associated RNA Transcripts (Mhrt or MyHEART)”. In the heart, the Brg1-Hdac-Parp chromatin repressor complex3 is activated by pathological stress to inhibit transcription of Mhrt. This stress-induced repression of Mhrt is important for the development of cardiomyopathy: however, Mhrt restoration at a pre-stress level protects the heart from hypertrophic cardiomyopathy. In humans, MHRT is also present at MYH7 loci and is found to be suppressing in many kinds of cardiomyopathies, suggesting a conserved lncRNA mechanism in human hypertrophic cardiomyopathy [83].

### 4.4. LncRNA-CAIF Attenuates HCM

Cardiac autophagy inhibitory factor (CAIF) is a unique long noncoding RNA that protects against myocardial infarction. The function of CAIF in end-stage cardiomyopathy was investigated by Wu et al. This research included patients with end-stage cardiomyopathy and the control subjects. CAIF was found to be downregulated in end-stage cardiomyopathy patients as compared to healthy controls, hence it may be used as a diagnostic and prognostic marker [84]. CAIF expression differentiated patients with end-stage cardiomyopathy from stable controls and predicted patient survival.

### 4.5. LncRNAs uc004cov.4 and uc022bqu.1 Mediate HCM

In another study, seven different cardiac and mitochondrial remodeling-related lncRNAs were quantified in control HOCM and HNCM patients. LncRNAs uc022bqu.1 and uc004cov.4 were substantially upregulated in patients with HOCM compared to controls, but no substantial deregulation was found in patients with HNCM. In both patients, the lncRNAs uc004coz.1, uc004cos.4, uc022bqw.1, uc011mfi.2, and uc022bqs.1 (LIPCAR) were not deregulated relative to the control group. When ROC curve analysis of the substantially increased lncRNAs in HOCM was conducted, both uc022bqu.1 and uc004cov.4 had the highest potential to considerably classify HOCM patients under the curve area of >0.68 [85]. 

### 4.6. High-throughput Screening of lncRNAs in HCM Patients

Hypertrophic cardiomyopathy (HCM) is one of the most common hereditary heart disorders. However, the signaling pathways and regulatory networks responsible for the manifestation of HCM are poorly understood. Recently, some bioinformatics studies have been done to profile the differentially expressed lncRNAs in HCM patients and to also understand the gene regulatory networks. Case in point, Guo et al. employed weighted correlation network analysis (WCNA) and linear models for microarray data (Limma) to analyze GSE68316 data from cardiac tissue in the Gene Expression Omnibus database. Their analysis did reveal that three circulating lncRNAs (lnc-P2RY6-1:1, ENST00000488040 and ENST00000588047) were significantly upregulated in the HCM cardiac tissue, but they didn’t elaborate how these circulating lncRNAs are involved in the molecular mechanism underlying the pathogenesis of HCM. Other similar studies have also been conducted. Readers can refer to these studies for details [86,87,88,89,90,91].

## 5. Probing the Transcriptome for Identification of lnc—RNAs Implicated in HCM

RNA-seq is an emerging technique in the transcriptome profiling system. Here we present an overview of experimental design and workflow that can be effectively adopted to generate a dedicated protocol, which can be used to identify the differentially expressed lncRNAs in HCM vs. normal heart tissues (Figure 3). 

The first and foremost step is the recruitment of HCM patients based on the results of genetic testing for HCM casual genes and collection of their heart tissue through myocardial biopsy. However, this can pose a challenge especially when it comes to obtaining ethical approval, in addition to the fact that myocardial biopsy is a specialized procedure that requires extensive training to perform. One of the ways to bypass this challenge will be to use heart tissue samples obtained from cadavers whose clinical history shows the presence of HCM. Our research group is in the process of standardizing a protocol for isolating total RNA from cadaveric heart tissue samples (Naidoo et al., unpublished data). Total RNA isolated from patients or cadaveric specimens can then be used for the generation of sequencing libraries and resulting libraries can then be sequenced in paired-end mode using a suitable platform such as the Illumina HiSeq system. 

The raw sequencing reads obtained can then be subjected to base quality filtering and adapter trimming. Clean reads thus obtained will then be required to align with the reference genome (Human-GRCh37) using a fast and sensitive alignment program for mapping next-generation sequencing reads. 

After assembling all clean paired-end reads, lncRNAs can be filtered according to specific defined criteria: (1) the assembled transcript must have specific strand information: (2) the length should be more than 200 bp: (3) it must have more than one exon and (4) lack of coding potential according to a defined coding potential calculator (CPC) algorithm, such that only the “noncoding” transcripts are obtained as an output. 

All long non-coding RNAs will then require incorporation in a differential expression analysis workflow (such as the Stringtie workflow freely accessible on the webpage of The Center for Computational Biology at Johns Hopkins University) leasing to the identification of differentially expressed genes (DEGs). A comparison of DEGs between normal and HCM heart tissues can then be performed using a bioinformatics tool such as sleuth (Readers are requested to refer to the excellent article: Differential analysis of gene regulation at transcript resolution with RNA-seq by Cole Trapnell, David G Henderickson, Martin Savageau, Loyal Goff, John L Rinn and Lior Pachter, Nature Biotechnology 31, 46–53 (2013); where this aspect has been discussed in detail.).

## 6. Conclusions

There has been considerable progression in the molecular mechanisms regarding hypertrophic cardiomyopathy over the past several decades. However, the pathological mechanisms of HCM for improved treatment strategies still require further elucidation. Covering a large body of evidence, this review illustrates that lncRNAs may act as an important participant in the complex network that regulates the pathological process of HCM, identifying them as potentially promising targets for the treatment of hypertrophic cardiomyopathy. It is an emerging avenue in providing new insights to explore the pathophysiologic mechanisms and establishing novel therapeutic targets for disease modulation and prevention.

## Figures and Tables

**Figure 1 ijms-22-09378-f001:**
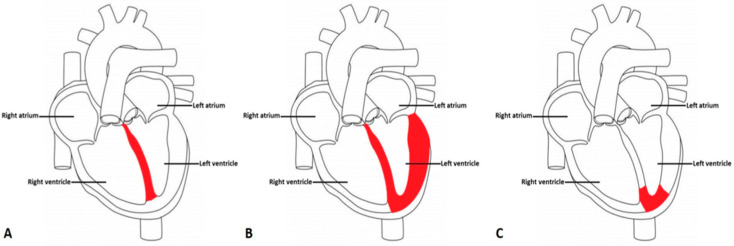
Variant Morphological Patterns of Hypertrophic Cardiomyopathy (**A**) Asymmetric Septal Hypertrophy with or without obstruction: asymmetric involvement of the interventricular septum where the stiffened, thickened area is shaded in red. (**B**) Symmetric or Concentric Hypertrophy: symmetrical stiffening and thickening of the left ventricle in a circumferential pattern shaded in red. (**C**) Apical Hypertrophy: stiffening and thickening of the apex of the left ventricle shaded in red.

**Figure 2 ijms-22-09378-f002:**
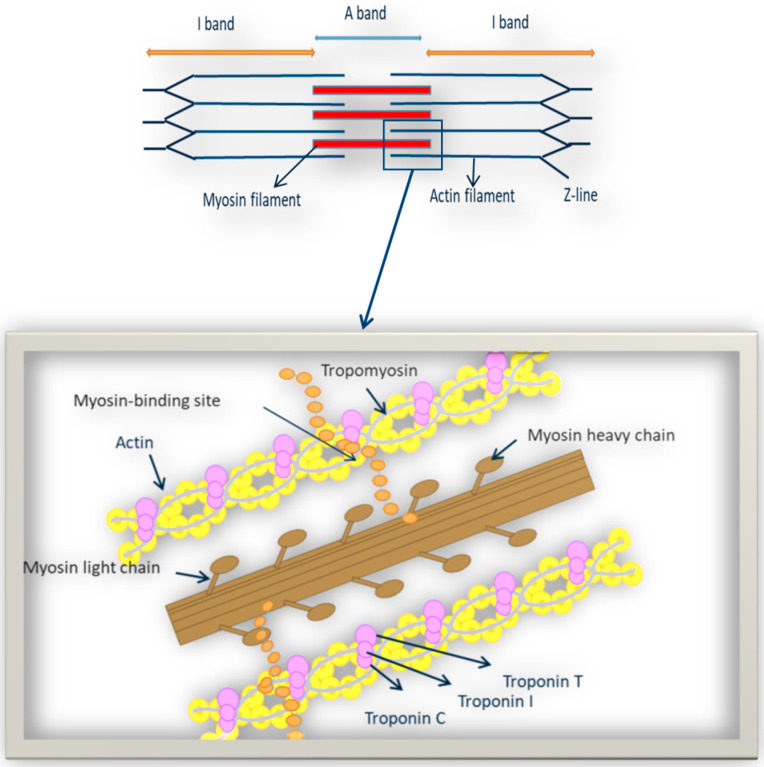
Representation of sarcomeric proteins responsible for expression of HCM.

**Figure 3 ijms-22-09378-f003:**
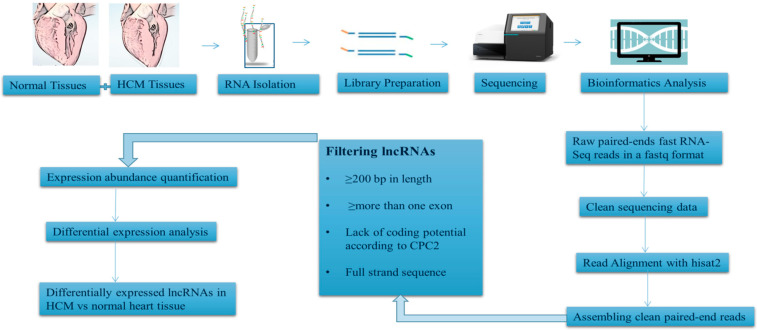
Schematic representation of experimental workflow involved in RNA-seq analysis of differentially expressed lncRNAs in normal and HCM patient’s sample.

**Table 1 ijms-22-09378-t001:** List of genes implicated in hypertrophic cardiomyopathy.

Gene	Protein	Location	Function	Locus	Frequency (%)
*ACTC1*	Cardiac actin-1	Thin Filament	Actomyosin interaction	15q14	<1
*TPMI*	α-tropomyosin		Places the troponin complex on cardiac actin	15q22.2	<1
*TNNI3*	Cardiac troponin I		Inhibitor of actomyosin interaction	19q13.4	<5
*TNNC1*	Cardiac troponin C		Calcium sensor in cardiac and slow skeletal muscle	3p21.1	<1
*TNNT2*	Cardiac troponin T		Regulator of actomyosin interaction	1q32.1	~10
*MYBPC3*	Myosin binding protein C	Thick Filament	Cardiac contraction	11p11.2	~40
*MYH7*	Cardiac myosin heavy chain β		ATPase activity, force generation	14q11.2	~40
*MYL2*	Cardiac myosin light chain 2		MYH7 binding protein	12q21.11	<1
*MYL3*	Cardiac myosin light chain 3		MYH7 binding protein	3p21.31	<1
*CSRP3*	Cysteine and glycine rich protein 3	Z-disk	Muscle LIM protein (MLP), a Z disk protein	11p15.1	<1
*MYOZ2*	Myozenin 2		Z disk protein	4q26	<1

**Table 2 ijms-22-09378-t002:** Phenocopies of hypertrophic cardiomyopathy.

Syndrome	Gene	Protein	Locus	Frequency
Danon’s syndrome	LAMP2	lysosome-associated membrane protein-2	Xq24	rare
Forbes disease (Glycogen Storage Disease Type 3)	AGL	Amylo-1,6-glucosidase	1p21	rare
Fabry’s disease (Lysosomal Storage Disorder)	GLA	a-Galactosidase A	Xq22	<5%
Pompe Disease (Glycogen Storage Disease Type 2)	GAA	a-1,4-glucosidase deficiency	17q25.2-q25.3	rare
Noonan syndrome/LEOPARD syndrome	PTPN11 SOS1 RAFI	v-Ki-ras2 Kirsten rat sarcoma viral oncogene Homolog Son of sevenless homolog 1 V-RAF-1 murine leukemia viral oncogene homolog 1	12q24.1 2p22-p21 3p25	rare rare
Friedreich’s ataxia	FXN	Frataxin	9q13	rare
Wolff-Parkinson-White syndrome	PRKAG2	AMP-activated protein kinase	7q35-q36.36	<1%
Barth syndrome/left ventricular noncompaction	DTNA TAZ	a-Dystrobrevin Tafazzin (G4.5)	18q12 Xq28	rare

**Table 3 ijms-22-09378-t003:** LncRNAs involved in pathogenesis of Cardiac Hypertrophy.

lncRNA	Mechanism of Action
Plscr4	Protective against cardiac hypertrophy by sponging miR-214 to derepress Mfn2 [46]
Chast	Promotes cardiac hypertrophy by activating NFAT signaling and downregulates Plekhm1 to induce cardiac remodeling processes [47]
CHRF	Induces cardiac hypertrophic response by sponging miR-489 to increase expression of Myd88, also by sponging miR-93 to disinhibit PI3K/Akt pathway [48]
MIAT	Contributing factor to the pathogenesis of cardiac hypertrophy and act by 1-Sponging miR-150 to increase expression of P300 [49,50] 2-Sponging miR-93 to activate PI3K/Akt/mTOR pathway via TLR4 [49]
Chaer	Increases pro-hypertrophic gene expression by interacting with PCR2 to disinhibit hypertrophic gene expression [51]
HOTAIR	Inhibits progress of cardiac hypertrophy by sponging miR-19 to derepress PTEN expression [52]
ROR	Promotes fetal genes and cardiomyocyte growth by sponging miR-133 [53]
MAGI1-IT1	Protects against cardiac hypertrophy by sponging miR-302e to derepress DKK1 and inactivate Wnt/beta-catenin signaling [54]
Meg3	Promotes cardiac hypertrophy by activating STAT3 to sponge miR-361-5p and derepress HDAC9 [55]
SYNE1-AS1	Promotes cardiac hypertrophy and activate SP1 to sponge miR-525-5p to derepress SP1, forming a positive feedback loop [56]
MALATI	Protects against cardiac hypertrophy by sponging miR-302e to derepress DKK1 and inactivate Wnt/beta-catenin signaling [57,58]
Ak045171	Promotes cardiac hypertrophy by binding with SP1, which promotes transcription activation of MEG3 [59]
TINCR	Attenuates cardiac hypertrophy by epigenetically silencing CaMKII [60]
UCA1	Promotes the progression of cardiac hypertrophy through competitively binding with miR-184 to enhance the expression of HOXA9 [61]
XIST	1-Regulates cardiac hypertrophy by sponging miR-101 to derepress TLR2 [62] 2-Attenuates cardiac hypertrophy by sponging miR-330-3p to derepress S100 [63]
TUG1	1-Contributes to cardiac hypertrophy via regulating miR-29b-3p [64] 2-Alleviates cardiac hypertrophy by targeting miR-34a/DKK1/Wnt-β-catenin signaling [65]
VDR/CASC15	Facilitates cardiac hypertrophy via miR-432-5p/TLR4 axis [66]
CYTOR	Protects against cardiac hypertrophy through miR-155 and downstream IKKi and NF-κB signaling [67]
FTX	Reduces hypertrophy by regulating the pten/pi3k/akt signaling pathway by sponging microrna-22 [54]
PVT1	Positively regulates cardiac hypertrophy by an unknown mechanism [68]
CHAR	Protects against cardiac hypertrophy via regulating miR-20b and the downstream PTEN/AKT pathway [69]
KCNQ1OT1	Attenuates cardiac hypertrophy through modulation of the miR-2054/AKT3 axis [70]
ZEB2-AS1	Stimulates cardiac hypertrophy by downregulating PTEN [71]
PEG10	Aggravates cardiac hypertrophy by positively regulating HOXA9 [72]
NEAT	Promotes cardiac hypertrophy through sponging microRNA-19a-3p/SMYD2 axis [73]
Ahit	Protects against cardiac hypertrophy through suz12 -mediated downregulation of mef2a [74]
Gm15834	Alleviates autophagy-induced myocardial hypertrophy by sponging miR-30b-3p [75]

**Table 4 ijms-22-09378-t004:** LncRNAs involved in pathogenesis of hypertrophic cardiomyopathy.

lncRNA	Mechanism of Action
Mhrt	Protect against pathological hypertrophic cardiomyopathy by inhibiting the Brg1-Hdac-Parp chromatin repressor complex to prohibit initiation of Myh6-to-Myh7 switch 2. Reduce myocardin acetylation/expression via HDAC5
MIAT	Contributing factor to the pathogenesis of hypertrophic cardiomyopathy and act by mediating the expression of miR-29a-3p
H19	Targets CaMKIIδ through miR-675 and functions as a negative regulator of hypertrophic cardiomyopathy
CAIF	Downregulated in end-stage cardiomyopathy
uc004cov.4	Upregulation leads to Hypertrophic obstructive cardiomyopathy-unknown mechanism
uc022bqu.1	Upregulation leads to Hypertrophic obstructive cardiomyopathy- unknown mechanism

## Data Availability

Not applicable.

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
