# Peer review of "Reconnoitering the Role of Long-Noncoding RNAs in Hypertrophic Cardiomyopathy: A Descriptive Review"

_ijms, 2021, doi:10.3390/ijms22179378_

Round 1

Reviewer 1 Report

The review is devoted to an interesting and important problem - the involvement of long noncoding RNAs (lncRNAs) in the development of hypertrophic cardiomyopathy (HCM). It is well structured and contains informative tables. 

However, it seems unjustified to include Section 5 “High-Throughput Screening of lncRNAs in HCM Patients” (line 235). A standard RNA-seq workflow for detecting differentially expressed RNAs when comparing abnormal and normal tissues is described in the text and depicted in Fig. 3. The text includes methodological details that are not characteristic of a descriptive review, such as "The heart tissues will be collected during septal myectomy and immediately placed in liquid nitrogen storage”. I would recommend that the authors exclude this section from the review.

Unfortunately, the overview is not exhaustive: it does not include all publications on the topic. Thus, when searching for the keywords "Hypertrophic cardiomyopathy AND lncRNA", PubMed offers 6 publications in 2020, but none of them is cited in the review. Authors should expand the list of works under consideration with publications that have appeared recently.

Author Response

PLEASE REFER TO THE ATTACHMENT FOR THE REVISED MANUSCRIPT FILE WHERE THE CHANGES IN RESPONSE TO THE REVIEWER 1 COMMENTS ARE SHOWN IN THE FORM OF TRACK CHANGES

Reviewer 1 Comments:

Comment 1: The review is devoted to an interesting and important problem - the involvement of long noncoding RNAs (lncRNAs) in the development of hypertrophic cardiomyopathy (HCM). It is well structured and contains informative tables. 

Authors’ response: We thank the reviewer for the positive and supportive comments.

Indeed, hypertrophic cardiomyopathy (HCM) represents one of the most common inherited cardiac ailments with an anticipated prevalence of 0.2 % in the general population. However, the molecular mechanisms that are implicated in the pathogenesis of HCM remain poorly defined.

Data from the Encyclopaedia of DNA Elements or the ENCODE project has effectively demonstrated that >90 % of the genome can be transcribed and the non-coding transcripts account for approximately 98 %. Typically, lncRNAs are involved in a variety of biological processes, and dysregulation of lncRNAs has been reported in numerous important human diseases, including cancer and neurological diseases such as schizophrenia. However, little is known about how dysregulation of lncRNAs may be responsible for HCM. This descriptive review aims to address this specific aspect, and therefore we firmly believe will be of immense value to researchers interested in understanding the underlying molecular mechanism of HCM.

Comment 2: However, it seems unjustified to include Section 5 “High-Throughput Screening of lncRNAs in HCM Patients” (line 235). A standard RNA-seq workflow for detecting differentially expressed RNAs when comparing abnormal and normal tissues is described in the text and depicted in Fig. 3. The text includes methodological details that are not characteristic of a descriptive review, such as "The heart tissues will be collected during septal myectomy and immediately placed in liquid nitrogen storage”. I would recommend that the authors exclude this section from the review.

Authors’ response: We thank the reviewer for the recommendation.

Our intention for including methodological details was to inform the reader with regards to a guide-plan that can be effectively implemented in investigating the role of lncRNAs in the causality of HCM.

We do concur with the reviewer that in depicting the guide-plan we included superfluous details, which detrimentally affected the flow and readability of the manuscript. In line, we have redrafted the guide-plan (removing specific details), which accompanies Figure – 3, in a way that it provides a comprehensive and general overview of a work-flow that can be easily and effectively adopted to design a protocol to investigate the role of lncRNAs in HCM. The redrafted section 5 now reads: “Here we present an overview of experimental design and workflow that can be effectively adopted to generate a dedicated protocol, which can be used to identify the differentially expressed lncRNAs in HCM vs normal heart tissues (Figure 3). The first and foremost step is recruitment of HCM patients based on the result of genetic testing for HCM casual genes and collection of their heart tissue through myocardial biopsy. However, this can pose a challenge especially when it comes to obtaining ethical approval, in addition to the fact that myocardial biopsy is a specialized procedure that requires extensive training to perform. One of the ways to bypass this challenge will be to use heart tissue samples obtained from cadavers whose clinical history shows the presence of HCM. Our research group is in the process of standardizing a protocol for isolating total RNA from cadaveric heart tissue samples (Naidoo et al, unpublished data). Total RNA isolated from patients or cadaveric specimens can then be used for the generation of sequencing libraries and resulting libraries can then be sequenced in paired-end mode using a suitable platform such as the Illumina HiSeq system.

The raw sequencing reads obtained can then be subjected to base quality filtering and adapter trimming. Clean reads thus obtained will then be required to be aligned to the reference genome (Human-GRCh37) using a fast and sensitive alignment program for mapping next-generation sequencing reads.

After assembling all clean paired-end reads, lncRNAs can be filtered according to specific defined criteria: (1) the assembled transcript must have specific strand information: (2) the length should be more than 200 bp: (3) it must have more than one exon and (4) lack of coding potential according to a defined coding potential calculator (CPC) algorithm, such that only the “noncoding” transcripts are obtained as an output.

All long non-coding RNAs will then require incorporation in a differential expression analysis workflow (such as the Stringtie workflow freely accessible on the webpage of The Center for Computational Biology at Johns Hopkins University) leasing to the identification of differentially expressed genes (DEGs). A comparison of DEGs between normal and HCM heart tissues can then be will be performed using a bioinformatics tool such as sleuth (Readers are requested to refer to the excellent article: Differential analysis of gene reg-ulation at transcript resolution with RNA-seq by Cole Trapnell, David G Henderickson, Martin Savageau, Loyal Goff, John L Rinn and Lior Pachter, Nature Biotechnology 31, 46–53 (2013); where this aspect has been discussed in detail.).

Comment 3: Unfortunately, the overview is not exhaustive: it does not include all publications on the topic. Thus, when searching for the keywords "Hypertrophic cardiomyopathy AND lncRNA", PubMed offers 6 publications in 2020, but none of them is cited in the review. Authors should expand the list of works under consideration with publications that have appeared recently.

Authors’ response: The authors thank the reviewer for the suggestion.

The focus of our manuscript is to review the mechanism of action of lncRNA in the development or prevention of HCM. Therefore, we didn’t focus on reviewing studies which primarily aimed at analysis of data obtained from public functional genomics data repository employing bioinformatics tools/workflows. Case in point, Guo et al employed weighted correlation network analysis (WCNA) and linear models for microarray data (Limma) to analyse GSE68316 data from cardiac tissue in the Gene Expression Omnibus database. Their analysis did reveal that three circulating lncRNAs (lnc-P2RY6-1:1, ENST00000488040 and ENST00000588047) were significantly upregulated in the HCM cardiac tissue, but they didn’t elaborate how these circulating lncRNAs are involved in the molecular mechanism underlying the pathogenesis of HCM. However, in the revised version of the manuscript we have alluded to this study and directed readers to other similar articles. The revised section depicted under Section 4.5 now reads “Hypertrophic cardiomyopathy (HCM) is one of the most common hereditary heart disorders. However, the signaling pathways and regulatory networks responsible for the manifestation of HCM are poorly understood. Recently, some bioinformatics studies have been done to profile the differentially expressed lncRNAs in HCM patients and some to understand the gene regulatory networks. Case in point, Guo et al employed weighted correlation network analysis (WCNA) and linear models for microarray data (Limma) to analyse GSE68316 data from cardiac tissue in the Gene Expression Omnibus database. Their analysis did reveal that three circulating lncRNAs (lnc-P2RY6-1:1, ENST00000488040 and ENST00000588047) were significantly upregulated in the HCM cardiac tissue, but they didn’t elaborate how these circulating lncRNAs are involved in the molecular mechanism underlying the pathogenesis of HCM. Other similar studies have also been conducted. Interested readers can refer to these studies for details. [86–91]”

We have also revised Table: 3 and included suitable references (Kindly refer to Reference 54 in the revised Table: 3).

Reviewer 2 Report

The present review aims at summarize the novel role of lcnRNAs in cardiac hypertrophy. During the past decade numerous studies highlighted the importance of lncRNAs in modulating cardiovascular cell signaling and in regulating the pathophysiological process of cardiac hypertrophy. Other papers have been already published in the field.

Authors should include the available literature and comment the previuos article in this field to differentiate their work from the previous ones ( J Cell Mol Med. 2020 Oct; 24(20): 11638–11645. doi: 10.1111/jcmm.15819; Front Physiol . 2019 Jan 29;10:30. doi: 10.3389/fphys.2019.00030; Front Pharmacol. 2020 Aug 21;11:1314. doi: 10.3389/fphar.2020.01314; Adv Exp Med Biol. 2020;1229:149-161. doi: 10.1007/978-981-15-1671-9_8; Heart Fail Rev. 2020 Nov;25(6):1037-1045. doi: 10.1007/s10741-019-09882-2)

Author Response

PLEASE REFER TO THE ATTACHED FILE FOR THE REVISED MANUSCRIPT WHERE THE REVISIONS ARE REFLECTED IN THE FORM OF TRACK CHANGES

Reviewer 2 comments:

Comment 1: The present review aims at summarizing the novel role of lcnRNAs in cardiac hypertrophy. During the past decade numerous studies highlighted the importance of lncRNAs in modulating cardiovascular cell signalling and in regulating the pathophysiological process of cardiac hypertrophy. Other papers have been already published in the field.

Authors’ response: We agree with the reviewer that other review articles have been published in the field. However, to our knowledge there is a dearth of review articles in the current literature that specifically focus on reviewing the mechanism of action of lnc-RNAs in development of hypertrophic cardiomyopathy.

We have addressed this gap in the current manuscript. We have also highlighted this aspect in the current manuscript through a sentence which reads “However, we focused on those studies only involving the known mechanism of action of lnc-RNAs in development of hypertrophic cardiomyopathy”.

Comment 2: Authors should include the available literature and comment the previuos article in this field to differentiate their work from the previous ones ( J Cell Mol Med. 2020 Oct; 24(20): 11638–11645. doi: 10.1111/jcmm.15819; Front Physiol . 2019 Jan 29;10:30. doi: 10.3389/fphys.2019.00030; Front Pharmacol. 2020 Aug 21;11:1314. doi: 10.3389/fphar.2020.01314; Adv Exp Med Biol. 2020;1229:149-161. doi: 10.1007/978-981-15-1671-9_8; Heart Fail Rev. 2020 Nov;25(6):1037-1045. doi: 10.1007/s10741-019-09882-2)

Authors’ response: The review articles recommended by the reviewer are related to the role of lncRNA in cardiac hypertrophy. Cardiac hypertrophy is not always associated with detrimental physiological outcomes. Case in point, exercise training-induced physiological cardiac hypertrophy also known as “athlete’s heart” presents cardioprotective effects and is not related to heart failure. We would like to direct the attention of the reviewer to the following article: Ooi JY, Bernardo BC, McMullen JR. The therapeutic potential of miRNAs regulated in settings of physiological cardiac hypertrophy. Future Med Chem. 2014 Feb;6(2):205-22. doi: 10.4155/fmc.13.196.” where this aspect has been dealt in detail.

The current review, however, focuses only on reviewing the role of lncRNA in hypertrophic cardiomyopathy. In line with is aim, all the published data obtained from the above indicated references on appraising lncRNAs’ role in hypertrophic cardiomyopathy is summarized in Table: 3 and has now been cited in a separate column. Also, the above-mentioned review articles are now referred to in section 4 of the review. The citations have been amended accordingly.  

Round 2

Reviewer 1 Report

The authors took into account all the comments of the reviewer.

Reviewer 2 Report

I am satisfied of the revised version of the manuscript